# Integrating Electronic Medical Records and Claims Data for Influenza Vaccine Research

**DOI:** 10.3390/vaccines10050727

**Published:** 2022-05-06

**Authors:** Constantina Boikos, Mahrukh Imran, Simon de Lusignan, Justin R. Ortiz, Peter A. Patriarca, James A. Mansi

**Affiliations:** 1Seqirus Inc., Kirkland, QC H9H 4M7, Canada; constantina.boikos@seqirus.com (C.B.); james.mansi@modernatx.com (J.A.M.); 2Nuffield Department of Primary Care Health Sciences, University of Oxford, Oxford OX2 6GG, UK; simon.delusignan@phc.ox.ac.uk; 3Center for Vaccine Development and Global Health, Baltimore, MD 21201, USA; jortiz@som.umaryland.edu; 4Immuno-Vax, LLC, Bethesda, MD 20814, USA; ppatriarca@immuno-vax.com

**Keywords:** influenza vaccines, citizen science, medical records systems, computerized, insurance, health, health information systems

## Abstract

Real-world evidence (RWE) increasingly informs public health and healthcare decisions worldwide. A large database has been created (“Integrated Dataset”) that integrates primary care electronic medical records with pharmacy and medical claims data on >123 million US patients since 2014. This article describes the components of the Integrated Dataset and evaluates its representativeness to the US population and its potential use in evaluating influenza vaccine effectiveness. Representativeness to the US population (2014–2019) was evaluated by comparison with demographic information from the 2019 US census and the National Ambulatory Medical Care Survey (NAMCS). Variables included in the Integrated Dataset were evaluated against World Health Organization (WHO) defined key and non-critical variables for evaluating influenza vaccine performance. The Integrated Dataset contains a variety of information, including demographic data, patient medical history, diagnoses, immunizations, and prescriptions. Distributions of most age categories and sex were comparable with the US Census and NAMCS populations. The Integrated Dataset was less diverse by race and ethnicity. Additionally, WHO key and non-critical variables for the estimation of influenza vaccine effectiveness are available in the Integrated Dataset. In summary, the Integrated Dataset is generally representative of the US population and contains key variables for the assessment of influenza vaccine effectiveness.

## 1. Introduction

There are many advantages of randomized controlled trials (RCTs) [1], including the distribution of known and unknown confounders between study groups through the randomization process. Despite this key strength, RCTs have substantial limitations. One of these is the trade-off between internal validity and external validity. Another is in balancing the establishment of a causal relationship (between a medical intervention and a specific outcome of interest) with the extent to which results are generalizable to other times, places, and populations [2,3,4]. RCTs are also typically time- and resource-intensive, which limits their feasibility in post-licensure evaluations of medical products [5].

The limitations of RCTs may, however, be addressed by real-world evidence (RWE) studies [6], generated through the analysis of real-world data. Real-world data relates to a patient’s health status and the healthcare received [7] and are collected from various sources, including electronic medical records (EMRs), medical and pharmacy insurance claims and billing activities, product and disease registries, and patient-generated information, as well as from other sources that can inform on health status, such as mobile devices [7].

RWE is particularly relevant in addressing questions relating to seasonal influenza vaccine performance. In particular, antigenic drift in circulating influenza viruses necessitate reformulation of influenza vaccines at least once a year, which makes it important to evaluate vaccine effectiveness annually. As such, RWE can provide timely insights into the safety and effectiveness of influenza vaccines under “real-world” conditions, and can be an important tool to inform and assess influenza vaccine policy [8,9,10,11], clinical practice guidelines [12], and product development [13,14].

Whilst vaccine efficacy is estimated by RCTs, vaccine effectiveness can be determined by a variety of observational study designs [15,16]. Claims data may be incomplete if clinicians do not bill all relevant diagnoses from a clinical encounter. Furthermore, billing is often time-intensive, and, in a fee-for-service setting, the payment a clinician receives for an office visit may not be directly related to the number or type of conditions for which they code. For these reasons, claims data alone are often imperfect reflections of the actual health status of a patient. On the other hand, EMR data contain clinical information for all patients receiving care by a clinician or medical practice [17], and allow the study of real-world clinical outcomes in near real time [18]. Moreover, EMRs contain an abundance of additional information, including laboratory results, vital signs, medical history, and demographics, as well as health problem lists [17,19]. While not without their limitations [17], the use of EMR data may significantly improve identification and understanding of health conditions without a coded diagnosis [19]. The linkage of EMR data with claims data leverages and improves the strengths of each individual data source [20]. Indeed, the value of combining evidence from multiple sources to improve estimation of vaccine effectiveness, including influenza vaccine effectiveness, has been highlighted by multiple global stakeholders [13,21,22,23,24]. Such integrated datasets provide a more comprehensive view of the patient experience within the healthcare system and clinical status [23].

This article describes a large dataset for influenza vaccine research that integrates de-identified EMR data from primary care with pharmacy and medical claims (hereby referred to as the Integrated Dataset) and reviews its applicability in generating RWE related to influenza vaccines. This article is intended to be a descriptive overview of the Integrated Dataset and not a formal fitness-for-use assessment of the dataset.

## 2. Materials and Methods

The Integrated Dataset is based on complete de-identified data from three ambulatory care EMRs (Veradigm Health Insights Ambulatory database) linked with both open and closed pharmacy and medical claims data (Komodo Healthcare Map) where available. The Integrated Dataset includes data since 2014 for a total of 123,229,120 US individuals, with information on 99,912,523 de-identified EMR patients with linked claims (open or closed) where available for 1 or more years, in addition to roughly 23,316,597 de-identified EMR patients with a partial year of linked claims data. Here, a “year of data” in the EMR is defined as at least one provider visit in a year, while in the linked data, a “year of data” is defined as at least one provider visit in the EMR data, as well as at least one claim or enrollment period in the same year. The Integrated Dataset is routinely updated with recent data; EMR data are available in almost real time, while claims data are available following a lag of several months to allow for adjudication and processing.

Three national primary care and specialty care EMR systems form the basis of the Integrated Dataset: Allscripts Professional, Allscripts Touchworks, and Practice Fusion. Systems include medical practices of a range of sizes, including small practices (1–3 physicians), medium-sized practices (4–40 physicians), and Integrated Delivery Networks (i.e., integrated networks of healthcare organizations). Separately, the Komodo Healthcare Map consists of anonymized patient-level US pharmacy and medical claims [25]. Patient encounter data for 150 million individuals in the Komodo Healthcare Map are derived directly from payer sources, including 100% fully integrated fee-for-service Medicare data, Medicare Advantage claims, commercial claims, and Medicaid claims.

Software for linkage of each individual de-identified dataset was provided by a third party (Datavant, San Francisco, CA, USA). Each individual dataset was required to meet the minimum protected health information (PHI) data requirements to ensure that linkage of the individual datasets was compliant with the HIPAA. The HIPAA of 1996 is a US federal law that required the creation of national standards to protect sensitive patient health information from being disclosed without the patient’s consent or knowledge. Deterministic matching algorithms were used to create two de-identified patient tokens from the identifiable information separately for each patient with records in the EMR and claims data sources. For patients in both sources with matches on both tokens, one unique patient identifier was created. The two data sources were then linked by the common patient identifier (Appendix A). The linked dataset was checked to verify that it contained no PHI and was evaluated and certified for HIPAA compliance by a third-party statistician. The linkage algorithm, as well as original EMR and claims datasets containing PHI, remain with the respective owners of the data and are not available for research purposes. Informed consent by patients is not necessary for use of the Integrated Dataset in research, as all records have been de-identified and integrated into real-world datasets. The ability to use patient-specific tokens allows for the linkage of other data elements to the Integrated Dataset.

Demographic information on geographic distribution, age, sex, race, and ethnicity of the entire population captured within the Integrated Dataset from 2014 to 2019 (*n* = 123,229,120) was compared with 2019 US Census Bureau data to evaluate the representativeness of the Integrated Dataset compared with the general US population [26]. As individuals captured within the Integrated Dataset must have medical insurance (such that their insurance claims are linked to their EMR data), the demographics within the Integrated Dataset were also compared with the 2016 National Ambulatory Medical Care Survey (NAMCS) data conducted by the US Centers for Disease Control and Prevention, which provides information on medical visits to office-based physicians and community health centers [27,28]. Summary statistics of key demographic variables (i.e., US geographic region, age, sex, race, and ethnicity) from these surveys were used for the comparison with the Integrated Dataset [28,29]. Furthermore, variables within the Integrated Dataset were compared with the key and non-critical variables for the design and interpretation of observational influenza vaccine studies proposed within the World Health Organization (WHO) guidance [9], using influenza-related medical encounters (defined as International Classification of Diseases, Tenth Revision (ICD-10) codes J09*–J11*) as the primary outcome (Table 1) [30].

**Table 1 vaccines-10-00727-t001:** Availability of variables within the Integrated Dataset for the evaluation of influenza vaccine effectiveness compared with WHO benchmarks [9].

Identified by the WHO [9]		Integrated Dataset
Key Variables	Rationale	
Influenza vaccines	Needed to identify vaccinated individuals	Available
Age	An important stratification factor for VE estimates, as VE may differ in different age groups Both vaccination coverage and risk of influenza virus infection vary by age	Available *Limitation: maximum age in variable is 89 years and artificially constrained in patients >89 years of age to remain compliant with HIPAA (i.e., to decrease the risk of potential re-identification of individuals)*
Sex	May be a strong variable related to healthcare utilization and vaccination in non-high-resource settings	Available
Race, ethnicity	Correlated with healthcare utilization in many parts of the world	Available
Date of symptom onset	Important variable for characterizing the influenza epidemic in the population: needed in cohort studies to calculate person-time at risk, and needed in case–control studies to sample controls (if using incidence-density sampling)	Available
Calendar time	Key variable in test-negative studies, because non-cases that are enrolled outside of an influenza season must be excluded from analyses to avoid bias Calendar time is also correlated with vaccine uptake and incidence of influenza, creating potential confounding by calendar time, although this confounding may not be meaningful in some settings	Available *The healthcare interactions captured within the primary care EMRs, pharmacy claims, and medical claims are tied to calendar time*
Time from symptom onset to specimen collection	May be associated with the sensitivity or specificity of influenza testing	*May be available within the EMR if testing for influenza was conducted within primary care*
Use of antivirals	Patients who have used antiviral medicines, either for treatment or for prophylaxis, are more likely to have false-negative test results; this can be used to exclude subjects from study enrollment	Available *Prescription for influenza antiviral medication available, although adherence is not*
**Non-critical Variables**		
Receipt of other vaccines (such as pneumococcal vaccines)	May be a marker for care-seeking behavior and/or propensity to seek influenza vaccination	Available
Prior history of influenza vaccination	Receipt of the prior year’s influenza vaccine may affect the effectiveness of the current season’s vaccine	Available
Presence and severity of cardiac or pulmonary comorbidities	Persons with chronic cardiac or pulmonary disease are at increased risk of influenza-associated complications if they are infected, and are therefore more likely to become cases in a hospital-based study In high-resource settings, underlying disease is also correlated with receipt of influenza vaccine, although in a non-linear fashion	Available *Information on hospitalization and reason for hospitalization (indicator of disease severity)*
Measure of outcome severity	Measures such as duration, subsequent hospitalization (particularly for outpatient outcomes), or death may be useful for assessing whether influenza vaccine reduces severity of outcomes in the vaccinated population (although this is complicated to estimate)	Available *Diagnosis information available for inpatients (potential marker of disease severity)*
Immunocompromising conditions	Generally, have been uncommon among subjects included in VE studies in high-resource settings and so have not been important confounders. However, in settings in which the prevalence of HIV/AIDS is high, HIV/AIDS may be an important confounder to measure	Available
Functional and cognitive limitations	Shown to be important confounders in VE studies among elderly adults in high-resource settings and particularly in relation to serious outcomes (i.e., hospitalization)	*Specific information on functional and cognitive limitations is not available; however, the construct of frailty may be generated using summary scores that leverage available information within the Integrated Dataset*
Access to medical care	Access to medical care will be population-dependent In some settings, availability and use of health insurance may affect patients’ ability to seek care at certain facilities	*Integrated Dataset represents individuals with health insurance; therefore, all individuals within the Integrated Dataset theoretically have access to medical care*
Socioeconomic status	Likely to be highly correlated with vaccination and with healthcare-seeking behavior	*Specific information on patient socioeconomic status not available within the Integrated Dataset*
Distance to study hospital/clinic	May be correlated both with access to vaccination and access to medical care	*Specific information not available within the Integrated Dataset as data evaluated retrospectively (no study hospital/clinic) and granular information on subject location of residence not available as per HIPAA requirements*

AIDS: acquired immune deficiency syndrome. EMR: electronic medical record. HIPAA: Health Insurance Portability and Accountability Act. HIV: human immunodeficiency virus. VE: vaccine effectiveness. WHO: World Health Organization.

## 3. Results

The Integrated Dataset is EMR-based and integrates information from pharmacy and medical claims data, where available, for a total capture of 123,229,120 individuals as of 2014. Of this total population, 63,830,391 (52%) have data on closed medical claims (i.e., patients for whom there are both claims and enrollment information), while the remaining individuals (*n* = 59,398,729) have claims but no enrollment information (i.e., open claims). Furthermore, the Integrated Dataset provides a longitudinal view of de-identified patients: one year of data (defined as at least one provider visit in the EMR data and at least one claim or enrollment period in the same year (either an open or closed claim)) is available for 56.5 million individuals, two consecutive years for 19.9 million individuals, three consecutive years for 9.4 million individuals, four consecutive years for 5.5 million individuals, five consecutive years for 3.4 million individuals, and five or more consecutive years for 5.2 million individuals.

Compared with 2019 US Census Bureau data, the population in the Integrated Dataset (2014–2019) provides a broad demographic representation of the US population (Figure 1A). Distributions of most age categories (Figure 1B) and sex (Figure 1C) are comparable, although the Integrated Dataset has a lower representation of individuals < 18 years of age (10.0 vs. 22.4%) and a higher representation of individuals > 65 years of age (30.0 vs. 16.0%) compared with the 2019 US census population. Over-representation of this older age group is likely driven by data sources that are built on the utilization of healthcare (i.e., EMR and open claims). The data sources should represent individuals who utilize healthcare that may differ from US Census information.

Compared with the general US population, the Integrated Dataset is less diverse by race (Asian: 0.9 vs. 5.6%; Black: 6.1 vs. 12.7%; White: 39.2 vs. 72.2%; other/unknown/missing: 53.4 vs. 9.5%; Figure 1D) and ethnicity (Hispanic/Latino: 7.1 vs. 15.9%; non-Hispanic/non-Latino: 74.1 vs. 84.1%; unknown/missing: 18.8 vs. 0.0%; Figure 1E). Of note, race/ethnicity data are derived solely from EMRs, where “unknown” or “not reported” are considered valid fields; patients and providers do not need to report patient race or ethnicity in EMRs.

Furthermore, there was general alignment between the population of the Integrated Dataset and the NAMCS survey with regard to geographic region (Figure 1A). Distributions across age categories (45–65 years and >65 years; Figure 1B) and sex (Figure 1C) are similar. Compared with the NAMCS survey populations of medical visits, the population within the Integrated Dataset is less diverse by race (Asian: 0.9 vs. 0.0–6.0%; Black: 6.1 vs. 10.6%; White: 39.2 vs. 83.8%; other/unknown/missing: 53.4 vs. 5.6%; Figure 1D) and ethnicity (Hispanic/Latino: 7.1 vs. 16.6%; non-Hispanic/non-Latino: 74.1 vs. 83.4%; unknown/missing: 18.8 vs. 0.0%; Figure 1E).

Influenza-related outcomes that may be evaluated include influenza-related medical encounters, defined as a medical visit with a recorded diagnosis of influenza disease defined by ICD diagnostic codes specific for influenza disease (ICD-10 J09*–J11*) [31,32]. The Integrated Dataset also provides information on influenza vaccination coverage by vaccine type, which is essential for brand-specific influenza vaccine effectiveness analyses. For example, during the 2018–2019 influenza season, 3.8 million individuals ≥ 65 years of age had a record of receiving the high-dose trivalent influenza vaccine, while 1.0 million and 0.9 million individuals received adjuvanted trivalent influenza vaccine and egg-based quadrivalent influenza vaccine, respectively. Furthermore, a larger number of individuals ≥ 4 years of age received egg-based quadrivalent influenza vaccine compared with cell-based quadrivalent influenza vaccine (8.0 vs. 2.0 million, respectively) during the 2018–2019 influenza season, predominantly in the 50–64-year age subgroup (2.7 vs. 0.8 million, respectively) (Table 2).

**Table 2 vaccines-10-00727-t002:** Number of individuals with a record of receiving an influenza vaccine, by vaccine type in the 2018–2019 US influenza season.

		Subjects ≥ 65 Years of Age ^†^ [33]		Subjects 4–17 Years of Age [34]		Subjects 18–49 Years of Age [34]		Subjects 50–64 Years of Age [34]		Subjects ≥ 65 Years of Age ^†^ [34]		Total Subjects ≥ 4 Years of Age [34]	
	aTIV	QIVe	HD-TIV	QIVc	QIVe	QIVc	QIVe	QIVc	QIVe	QIVc	QIVe	QIVc	QIVe
2018–2019 influenza season	1,031,145	915,380	3,809,601	78,602	1,628,038	700,729	2,641,268	828,460	2,743,654	517,639	987,943	2,125,430	8,000,903

^†^ The number of individuals receiving QIVe in the ≥65 years of age subgroup during this season represents two separate analysis cohorts. aTIV: adjuvanted trivalent influenza vaccine. HD-TIV: high-dose trivalent influenza vaccine. QIVc: cell culture-based quadrivalent influenza vaccine. QIVe: egg-based quadrivalent influenza vaccine.

With respect to the generation of RWE for influenza vaccine effectiveness studies, the principal WHO’s key variables, as well as many of the non-critical variables, for the estimation of influenza vaccine effectiveness are available in the Integrated Dataset, including the commonly evaluated variables used by the US Centers for Disease Control in influenza vaccine effectiveness studies [35] (Table 1). Specifically, information on the following variables, identified as key by the WHO, can be ascertained from the Integrated Dataset: influenza vaccines (by brand/type), calendar time, prescription of medications (such as influenza antivirals), and key demographic variables (age, sex, race, ethnicity, and geographic region). Other non-critical variables identified by the WHO can also be ascertained from the Integrated Dataset, including information on the receipt of non-influenza vaccines (such as pneumococcal vaccines), prior history of influenza vaccination, presence and severity of cardiac or pulmonary comorbidities (information on hospitalization and reason for hospitalization as an indicator for disease severity), measure of outcome severity (diagnosis information as a potential marker for disease severity), and immunocompromising conditions. Some variables identified by the WHO are not readily available within the Integrated Dataset. For example, a key variable identified by the WHO is the time from influenza symptom onset to laboratory specimen collection. In addition, information on prescription of medications (key variable), including influenza antivirals, is also available within the Integrated Dataset. However, as with many study designs (often including RCTs), confirmation of patient adherence to treatment is not possible from data in the Integrated Dataset. Moreover, direct measures of functional limitations (non-critical variable), such as frailty in older adults, are neither specifically nor systematically available within the Integrated Dataset. However, these can be estimated using proxy measures derived from diagnostic codes for conditions that impact activities of daily living, which would be captured in the Integrated Dataset [36]. Information on other non-critical variables, such as a patient’s physical proximity to the hospital or clinic, as well as specific measures of socioeconomic status (SES), is not directly available from the Integrated Dataset; detailed information on patient location of residence is not available in order to achieve compliance with HIPAA. Nonetheless, in a similar way to the construct of frailty, it is possible to generate proxy measures of SES using diagnostic codes for proxy measures of SES available within the Integrated Dataset [31].

## 4. Discussion

The Integrated Dataset is a distinct, US population-based repository of real-world data that integrates medical information from primary care EMRs with pharmacy and medical claims information, where available, for a total capture of 123,229,120 individuals as of 2014. Patient demographic information, medical history, outpatient and inpatient medical diagnoses, hospitalizations, immunizations, medication prescriptions, and most key and non-critical variables identified by the WHO as key for influenza vaccine research are all available within the Integrated Dataset [9]. An advantage of the Integrated Dataset will be its ability to link demographic information, vaccine exposure, clinical diagnosis of influenza disease (defined using ICD-10 codes J09*–J11*, which are specific for influenza disease [37]), and confounding variables necessary for influenza vaccine effectiveness analyses. Furthermore, the Integrated Dataset can be used to conduct retrospective analyses (i.e., cohort, case–control, and other case-based retrospective designs, such as the self-controlled case series design) [38]. Evaluations of relative vaccine effectiveness are relatively unaffected by exposure misclassification, as the exposure groups are ascertained based on the presence of specific codes indicating the receipt of a vaccine. Appropriate analytical methods that may be implemented include standard commonly used approaches, such as univariate, multivariate, and stratified regression analyses [32,39]. The large volume and variety of information (i.e., variables) in the Integrated Dataset also allow for the use of more sophisticated adjustment methodologies, including propensity score approaches (e.g., high-dimensional propensity scores and inverse probability of treatment weighting) [40,41].

Data within the Integrated Dataset are updated periodically to ensure capture of recent EMRs and claims. Protection of healthcare information from human subjects in the Integrated Dataset is maintained by meeting HIPAA guidelines, ensuring that the confidentiality and security of PHI remain as the patient records and claims are de-identified, aggregated across practices or payers, and linked to the Integrated Dataset. Although the US Food and Drug Administration has issued guidance on the use of EMRs in clinical investigations, requirements for further guidance are being considered around using EMRs to generate RWE [13]. Moreover, the Integrated Dataset may be expanded by the addition of data components through further linkage and de-identification. Whilst the Integrated Dataset is not the only large dataset available for RWE research [24,42,43]; it is one of the first, and largest, datasets integrating EMR and claims data for influenza vaccine research.

The representativeness of the Integrated Dataset to the US population is suggested by the alignment of key demographic variables to the 2019 US Census Bureau data [26], as well as NAMCS and National Hospital Ambulatory Medical Care Survey data [28,29]. In practice, the ability of the Integrated Dataset to be used in the evaluation of influenza vaccine effectiveness has already been demonstrated in several published analyses [33,34,44,45,46]. In addition, estimates of relative vaccine effectiveness generated using cohorts from the Integrated Dataset were included in a systematic review [47]. This review evaluated RWE for the MF59-adjuvanted trivalent/quadrivalent influenza vaccines (aTIV/aQIV) from non-interventional studies, published from 1997 through to 15 July 2020. Vaccine effectiveness estimated from cohort studies conducted in the same influenza seasons was generally comparable in magnitude or had overlapping confidence intervals compared with estimates from retrospective cohort studies conducted using the Integrated Dataset. Between-study variability in the effect estimates is likely attributable to differences in the underlying study populations and the study outcome definitions used [47].

A limitation of the Integrated Dataset is the lack of specific information on length of hospitalization, which is contingent on discharge date and not systematically reported in patient records. Furthermore, a larger proportion of data on race and ethnicity is missing (unreported/unavailable) in the Integrated Dataset versus the US Census Bureau and NAMCS data. This is a result of information on patient race or ethnicity not being included in the claims data and patient de-identification into the four race categories to protect patient privacy. This may be due, in part, to either the removal of information in order for the Integrated Dataset to comply with HIPAA requirements, or patients not providing this information. Regarding the variables identified by the WHO for the generation of RWE for influenza vaccine effectiveness studies, time from influenza symptom onset to laboratory specimen collection, a key variable, may not be systematically available within the Integrated Dataset, as this information is dependent on healthcare providers recording the date of symptom onset. This patient information and data were also omitted from key RWE studies [13,14]. Moreover, some non-critical variables, such as information on functional and cognitive limitations, are not systematically available within the Integrated Dataset, but can be estimated using proxy measures derived from other conditions captured. Other non-critical variables, including access to medical care, distance to a hospital or clinic, and SES, are not available within the Integrated Dataset in order to achieve compliance with HIPAA. Nevertheless, most key and non-critical variables identified by the WHO are available in the Integrated Dataset, demonstrating its value for the generation of RWE for influenza vaccines.

The WHO guidance on the evaluation of influenza vaccine effectiveness includes influenza-like illness as an outcome of interest, noting that although vaccine effectiveness against influenza-like illness will be lower than vaccine effectiveness against laboratory-confirmed influenza outcomes, the extent of this underestimation will vary depending on non-influenza causes of influenza-like illness within the study [9]. As the influenza-related outcomes in the current Integrated Dataset relate to diagnosis of disease using ICD codes rather than on a specific constellation of symptoms (as with influenza-like illness), it is possible that the vaccine effectiveness estimates generated may be an underestimation. Although information on laboratory diagnosis of influenza may be present in patient EMRs, important data regarding the diagnostic accuracy of the laboratory test used is unlikely to be available. It should be noted, however, that the use of restricted time periods for assessment of influenza, as recommended by the WHO, allows maximization of the proportion of non-specific outcomes that are caused by influenza [9]. Positive influenza tests reported to the CDC by public health laboratories have previously been shown to overlap with incidence of influenza-related medical encounters, supporting the use of ICD-10 codes (J09*–J11*) in evaluations of influenza [44,45]; ICD-10 codes have demonstrated a high positive predictive value (96%) with laboratory-confirmed influenza [31]. In addition, data contained in the Integrated Database are derived from patients who attend medical practices that provide data to the Veradigm Health Insights Ambulatory databases, and who also have claims captured in Komodo’s Healthcare Map. Additionally, the availability of information in EMR data may be confounded by underlying diagnoses potentially adding bias to our results [17,19]. Nonetheless, as reported above, the population of the Integrated Dataset is similar to the general population in the US, and its ability to be used for influenza vaccine research has been demonstrated [33,34,45]. As the Integrated Dataset only includes patients with fully private insurance coverage, the findings are most applicable to insured individuals in the US who are eligible to receive an influenza vaccine.

## 5. Conclusions

The Integrated Dataset combines data from EMRs with pharmacy and medical claims for a population of more than 123 million individuals. A larger proportion of data on race and ethnicity are missing (unreported/unavailable) in the Integrated Dataset compared to the US Census Bureau and NAMCS data. However, the Integrated Dataset is generally representative of the US insured population and, in previous studies, has demonstrated its use in generating RWE; this could help inform public health, clinical, and regulatory stakeholders on influenza vaccine performance.

## Figures and Tables

**Figure 1 vaccines-10-00727-f001:**
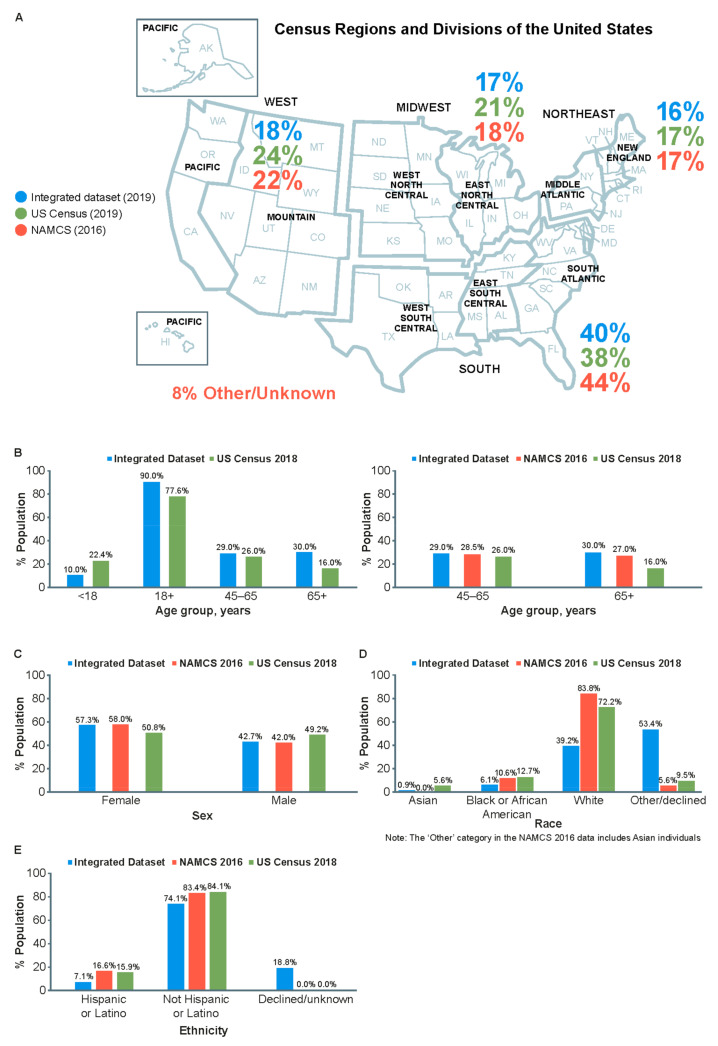
Geographic distribution of all subjects (*n* = 123,229,120, representing the full integrated dataset of EMR and both open and closed medical claims from 2014 to 2019) (**A**) and distribution by age (**B**), sex (**C**), race (**D**), and ethnicity (**E**) within the Integrated Dataset versus 2018 US Census and 2016 NAMCS data. EMR: electronic medical record. NAMCS: National Ambulatory Medical Care Survey.

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
