# Peer review of "Integrating Electronic Medical Records and Claims Data for Influenza Vaccine Research"

_vaccines, 2022, doi:10.3390/vaccines10050727_

Round 1

Reviewer 1 Report

The authors evaluated the usefulness of a large database, derived from an operation of integration of primary care electronic records with pharmacy and medical claims data, from more than one hundred million US citizens. They found that most of the key and non-critical variables suggested by WHO, for the estimation of influenza vaccine effectiveness were available, concluding that kind of data information aggregation may be very useful for different public health stakeholders.

Reviewer 2 Report

This article provides introductory descriptive information that would help other researchers decide if the Integrated Dataset might be useful for their particular project.

Regarding Table 1, in some cases it is not clear to which Key Variable the Rationale and Integrated Dataset comments apply.  I suggest placing horizontal lines across the three columns to separate the material associated with each Key Variable from the material applicable to the Key Variable listed above or below.

Reviewer 3 Report

In the current manuscript, Boikos et al explore the utility of electronic medical records for evaluation of the effectiveness of vaccines against influenza A virus.  

A strength of the current manuscript is the ability to correlate vaccine uptake with later clinical outcomes available in medical records.  

A major limitation of the current study is the lack of evaluation and presentation of data regarding vaccine effectiveness using electronic medical records.  It is unclear why In lieu of actually evaluating vaccine effectiveness with electronic medical records the authors simply state the information is available.   This is essential to significantly advance in the field.

Other concerns regarding the manuscript in its current form are as follows:

  1.  The introduction of the manuscript is long and would benefit from shortening.  
  2. How privacy issues will be ensured should be touched upon.
  3. A number of elements in Figure one are not legible in the color scheme chosen by the authors when the document is printed.   The fonts in B to E are too small to read easily.

Reviewer 4 Report

Integrating Electronic Medical Records and Claims Data for Influenza Vaccine Research

Constantina Boikos et al.

General:

The paper describes the construction of an integrated dataset where primary care electronic medical records are linked to pharmacy and medical claims yielding a real world evidence dataset. The data available in the integrated dataset were evaluated for its utility in the evaluation of influenza vaccines. The paper is clearly written and the results presented show the potential utility of the integrated dataset for the evaluation of influenza vaccines.

Specific comments:

Lines 75-76: The additional information that can be obtained from the EMRs are likely confounded by underlying diagnoses and may thus cause some degree of bias. This should be addressed in the discussion.

Lines 95-96: It would be interesting to know the distribution of the years of data available for the integrated dataset.

Discussion: in lines 332-335 it is mentioned that the influenza ICD codes have a high positive predicted  value with laboratory-confirmed influenza. Although this statement is obviously true, it is very likely that a laboratory confirmation is sought when and ICD influenza code is used. I’m not sure whether all influenza infections will get an ICD influenza code, particularly when the presentation is atypical (e.g. in the elderly).

Round 2

Reviewer 3 Report

The reviewer thanks the authors for their responsiveness to comments.

In order to accurately reflect the work presented in the paper, alternate phrasing for 'demonstrating the utility' of RWE is needed throughout the manuscript.  The later phraseology implies an 'ability to perform a function'.  Without inclusion of results to support that RWE can be used to evaluate vaccine effectiveness, for example from past years where vaccine effectiveness in known,  RWE analysis simply has potential as a resource tool, but needs to be validated with actual data.  

This is a major concern that has been brought up by past and current review, which should be addressed. 

Author Response

Reviewer Comment

Response

Location of Reviewer Comment

Reviewer 3

In order to accurately reflect the work presented in the paper, alternate phrasing for demonstrating the utility' of RWE is needed throughout the manuscript. The later phraseology implies an 'ability to perform a function'. Without inclusion of results to support that RWE can be used to evaluate vaccine effectiveness, for example from past years where vaccine effectiveness in known, RWE analysis simply has potential as a resource tool, but needs to be validated with actual data.

This is a major concern that has been brought up by past and current review, which should be addressed.

We have amended the manuscript to omit any phrasing including and similar to ‘demonstrating the utility’. Instances of this have been found in:

·        Abstract, line 30

·        Discussion, lines 237–245

·        Conclusion, lines 296

Global comment
